# Anti-Oxidative and Anti-Aging Activities of Porcine By-Product Collagen Hydrolysates Produced by Commercial Proteases: Effect of Hydrolysis and Ultrafiltration

**DOI:** 10.3390/molecules24061104

**Published:** 2019-03-20

**Authors:** Geun-Pyo Hong, Sang-Gi Min, Yeon-Ji Jo

**Affiliations:** 1Department of Food Science and Biotechnology, Sejong University, 209 Neungdong-ro, Seoul 05006, Korea; gphong@sejong.ac.kr; 2Animal Resources Research Center, Konkuk University, 120 Neungdong-ro, Seoul 05029, Korea; minsg@konkuk.ac.kr; 3Department of Agriculture, Food and Nutritional Science, University of Alberta, 116 St. and Ave., Edmonton, AB T6G 2P5, Canada

**Keywords:** hydrolysis, commercial proteases, collagen hydrolysate, antioxidant, anti-aging

## Abstract

To investigate methods for improving the processing of porcine waste, porcine skin was hydrolyzed using different commercially available proteases (Alcalase, Flavorzyme, Neutrase, Bromeline, Protamex, and Papain) under several optimal conditions. Following enzymatic hydrolysis, the collagen hydrolysates (CHs) were fractionated by molecular weight (3 kDa) via membrane ultrafiltration. The CHs were analyzed for physical properties (pH, protein recovery, free amino group content, molecular weight distribution, and amino composition) as well as for functional properties (antioxidant activities and anti-aging activities). Among the CHs, CHs hydrolyzed by Alcalase (CH-Alcalase) exhibited the highest degree of hydrolysis compared to other CHs. Both “CH-Alcalase” and “CH-Alcalase < 3 kDa” fractions showed a considerably high antioxidant activity and collagenase inhibition activity. Therefore, resulting bioactives have potential for development as antioxidants and anti-aging ingredients in the food, cosmetics, and pharmaceuticals, from animal by-products.

## 1. Introduction

With increasing life expectancy, skin anti-aging procedures are receiving much attention. Collagen, used in skin care products as a cosmeceutical ingredient for anti-aging, is widely used in the cosmetic market. Collagen is a high molecular weight, fibrous protein of animal origin, found abundantly in connective tissue, skin, tendons, cartilage, ligaments, teeth, nails, and hair of humans and animals. Livestock remains the main source of industrial collagen [1]. Annually, large quantities of livestock by-products are discarded worldwide, as waste, by the food and meat processing industries. However, these may be utilized as important protein resources for novel food materials or converted to value-added products via hydrolysis, which is widely applied to improve and upgrade functional and nutritional properties of proteins [2,3]. Collagen is also increasingly being considered as a source of bioactive peptides, which have shown promise as beneficial compounds for use in nutritional or pharmaceutical applications. Therefore, hydrolysis of collagen may play an important role in improving its bioavailability and suitability for use in various commercial processes. 

Physiological activities of bioactive peptides, which usually contain 2–20 amino acid residues (<6000 Da), are based on the composition and sequence of their amino acids [2,4]. Due to their structural properties such as molecular weight, bioactive peptides may possess specific characteristics that affect numerous physiological processes related to antimicrobial, antioxidant, emulsifying, antihypertensive, antidiabetic, and immunomodulatory functions in organisms [1,5,6]. Generally, low molecular weight hydrolysates display lower viscosity, better dispersion, higher hydrophobicity, and smaller particle size [7]. The degree of hydrolysis directly affects the molecular weight and amino acid composition of the hydrolysates [8]. Therefore, the hydrolysis process may be necessary for producing novel bioactive peptides.

Protein hydrolysis, leading to the cleavage of peptide bonds, may be carried out via enzymatic or chemical processes [9]. Chemical processes, including alkaline or acid hydrolysis using solvent extractions, are harmful not only to the environment but also to humans who consume the resulting products [4,10]. Chemical processes, which are difficult to control, yield products containing modified amino acids [9]. In contrast, enzymatic hydrolysis may be performed under mild conditions, thus avoiding extreme environments that are required by chemical treatments. Moreover, the processes neither produce side reactions nor decrease the nutritional value of the protein source [8]. Several proteases, including Alcalase, pepsin, Protamex, trypsin, and Neutrase, are commonly used to hydrolyze proteins, resulting in various types of protein hydrolysates and peptides with a variety of bio-functional activities. One study reported producing collagen hydrolysates with antioxidant activity from porcine skin gelatin using enzymatic hydrolysis by pepsin and pancreatin [11]. Salmon by-products were hydrolyzed to produce peptic hydrolysates using various proteases, where the final peptides showed excellent antioxidant and anti-inflammatory properties [12].

The objective of the research was: (a) to identify the most active collagen hydrolysates produced by enzymatic hydrolysis using different commercially available proteases (Alcalase, Flavorzyme, Neutrase, Protamex, Bromeline, and Papain); (b) to confirm the fractionation effect of collagen hydrolysates obtained from enzymatic hydrolysis; and (c) to determine their in vitro antioxidant and anti-aging activities.

## 2. Results and Discussion

### 2.1. Enzymatic Hydrolysis Effect on Collagen Hydrolysates

#### 2.1.1. pH

Functional properties of proteins may be enhanced through hydrolysis by certain proteases. In this study, six different proteases (Alcalase, Flavorzyme, Neutrase, Bromeline, Protamex, and Papain) were used to hydrolyze collagen from porcine skin in order to obtain active collagen hydrolysates with low molecular weights. The change of pH in collagen hydrolysates (CH) is shown in Figure 1. The pH of the collagen suspension (5% porcine skin mixture) was initially 6.39–6.55 (at 0 h). CH-Alcalase and CH-Flavorzyme reached optimal pH (pH 8.0) at 12 h of incubation. CH- Protamex, CH-Bromeline, and CH-Papain reached optimal pH (pH 7.0) at 6 h of incubation, whereas CH-Flavorzyme continued to decrease from pH 6.39 to pH 5.69 for 24 h incubation (Figure 1). The pH of a protein hydrolysate is an important factor that regulates enzymatic hydrolysis reactions [13]. Therefore, the generation of collagen hydrolysates under conditions of differing pH may be due to pH-dependent changes in enzyme conformation, which may impact their bio-activation and physicochemical properties.

#### 2.1.2. Sodium Dodecyl Sulfate-Polyacrylamide Gel Electrophoresis (SDS-PAGE)

Sodium dodecyl sulfate-polyacrylamide gel electrophoresis (SDS-PAGE) profiles of CHs treated with various proteases for 24 h are shown (Figure 2). The peptide profile of the collagen suspension (at 0 h) showed a visible band in the low-to-high molecular weight range. Results indicated rapid hydrolysis in CH-Alcalase, CH-Neutrase, CH-Bromeline, and CH-Protamex, as there were no bands after 3 h incubation. The high molecular weight range bands of CH-Alcalase, CH-Neutrase, and CH-Protamex decreased in a time-dependent incubation manner. However, CH-Bromeline did not vary with incubation time because it has been completely hydrolyzed after just 1 h incubation. Interestingly, the CHs completely disappeared when the sample reacted with Alcalase or Neutrase for 6 h, indicating that collagen acted as a suitable substrate for some of the other enzymes tested. However, bands of CH-Flavorzyme and CH-Papain appeared in the low-to-high molecular weight range after 24 h. Proteins can be partly hydrolyzed into peptides or amino acids by enzymatic hydrolysis, depending on protease type, incubation time, and the protease-to-substrate ratio (protein amount) [14]. The enzymatic hydrolysis process is mainly determined by whether the protease type involved is an endopeptidase or an exopeptidase [15,16]. Since Alcalase, Neutrase and Protamex are endopeptidases, their hydrolysis processes may be expected to break peptide bonds from non-terminal amino acids randomly, thereby facilitating further hydrolysis of proteins. On the other hand, Flavorzyme is both an endo- and exopeptidase, which breaks the non- or N-terminal of peptide chains. In this study, collagen protein contained binding sites, which were more susceptible to exopeptidase activity than to endopeptidase activity of Flavorzyme. Thus, due to exopeptidase activity, hydrolysis of collagen proteins by Flavorzyme involved step-by-step breaking of peptide bonds from the N-terminal of amino acids, which decelerated protein hydrolysis.

#### 2.1.3. Protein Recovery and Free Amino Group Content

Protein recovery of the CHs obtained via different commercial proteases is shown (Figure 3A). The control (collagen suspension) had a protein recovery content of 1.19 mg/mL. Protein recovery is a parameter of the efficiency of enzymatic hydrolysis. Low protein recovery content of CH is indicative of increased protein breakdown. Furthermore, protein recovery correlates well with protein solubility. Higher hydrolytic activity may be caused by an increase in exposed hydrophobic side chains, leading to increased hydrophobic interactions between proteins and/or peptides. These hydrophobic side chains may lead to decreased protein solubility, thus resulting in increased protein recovery of hydrolysates. In our study, protein recovery content was well-correlated with changes in the SDS-PAGE patterns. The lowest and highest protein recovery content levels were recorded for Alcalase and Flavorzyme hydrolysis, respectively. For hydrolysis efficiency related to incubation time, CH-Alcalase, CH-Bromeline and CH-Protamex drastically decreased within one hour of incubation compared to the control. The decrease in protein recovery content resulting from protein hydrolysis showed an inverse relationship with the amino group content. Furthermore, most CHs, except for CH-Neutrase, exhibited low protein content between 3 h and 12 h. After 24 h incubation, their protein content increased again in CH-Alcalase, CH-Flavorzyme, CH-Neutrase, CH-Bromeline, CH-Protamex, and CH-Papain as follows: 0.83 mg/mL, 1.29 mg/mL, 0.75 mg/mL, 0.97 mg/mL, 0.77 mg/mL, and 1.11 mg/mL, respectively. Some reports indicate that the optimum hydrolysis time depends on several factors such as the species selected, the raw material, the enzyme used, the raw material/enzyme rate, the concentration of the enzyme, the degree of hydrolysis pursued, etc. [16]. Some reports indicate that extended enzymatic hydrolysis (over 24 h) may lead to a reduction in protease activity; thus, enzymatic hydrolysis should be carried out within 24 h of enzymatic hydrolysis [17,18].

The free amino group content of CH, which is an indicator of enzymatic hydrolysis efficiency, was estimated (Figure 3B). The expected outcome of protein breakdown is that protein may be hydrolyzed into shorter peptide products. The free amino group content of the control was initially 0.14 mg/mL. The free amino group content of all CHs dramatically increased with increasing incubation time. After 24 h incubation, the free amino group content by descending order was: CH-Alcalase (1.00 mg/mL) > CH-Protamex (0.70 mg/mL) > CH-Neutrase (0.68 mg/mL) > CH-Bromeline (0.67 mg/mL) > CH-Flavorzyme (0.53 mg/mL) > CH-Papain (0.40 mg/mL). Commonly, differences in total protein or free amino group content of enzymatically hydrolyzed samples may be related to enzyme specificity. It is dependent on the properties of an enzyme during the enzymatic hydrolysis process [16,17]. For example, alkaline proteases (such as Alcalase) exhibit higher hydrolytic activities compared to acidic or neutral proteases.

### 2.2. Ultrafiltration Effect on Collagen Hydrolysate Properties

An ultrafiltration process may be a useful, industrially advantageous method for producing small peptide fractions with a desired molecular size and high bioactivity, depending on the composition of the starting hydrolysate and the activity being studied [19]. Solubility, free amino group content, and the yield of CHs after lyophilization is shown (Table 1). The solubility and free amino group content of the control were 11.95% and 0.79%, respectively. After enzymatic hydrolysis and ultrafiltration with 3 kDa molecular weight cut-off, the highest solubility (21.17%) and free amino group content (14.17%) were observed in CH-Alcalase < 3 kDa. However, CH-Alcalase < 3 kDa was the lowest yield (12.42%) observed. Therefore, although ultrafiltration is useful in separating CHs with low molecular weight, it may cause a reduction in yield.

The average molecular weight of protein hydrolysates is an important factor that determines their biological properties [19]. Generally, an average fraction with MW < 3 kDa represents a collagen hydrolysate; an average fraction with MW > 50 kDa represents gelatin; and an average fraction with MW > 300 kDa represents collagen [20,21]. The relative molecular weight distribution of the control (pretreatment sample), CH-Alcalase, and CH-Alcalase < 3 kDa is depicted in Figure 4. The molecular weight distribution was over 20,100 Da for the control, which did not include the collagen hydrolysate (MW < 3 kDa). This could not be numerically provided in this study, as the detection limit of the index detector system only ranged from 106 to 20,100 Da. However, CH-Alcalase showed detectable values in a higher range of relative molecular weight distribution, which ranged from 20,100 Da to 4270 Da (maximum peak: 12,600 Da). In CH-Alcalase < 3 kDa, the molecular weight distribution mainly showed three peaks: one with a MW of approximately 4270 Da (maximum peak), one with a MW of approximately 424 Da, and one with a MW of approximately 222 Da and 102 Da. Ultrafiltration is an effective purification method used to obtain low molecular weight peptides from crude hydrolysates. Results indicated that enzymatic hydrolysis by Alcalase clearly reduced the high MW of the control (either collagen or gelatin), and that ultrafiltration was an effective purification method that can be used to obtain low molecular weight peptides (<3 kDa) from crude collagen hydrolysates. Reportedly, low molecular weight peptides (2–20 amino acids) are more biologically active compared to their parent polypeptide/proteins, which are larger [22]. 

Amino acid composition in the CHs is shown (Table 2). The CHs by different proteases had different amino acid compositions and antioxidant properties [23]. The amino acid composition of collagen was rich in glycine (Gly), proline (Pro), and glutamic acid (Glu). The amino acid content of CHs (CH-Alcalase and the CH-Alcalase < 3 kDa) increased more than that of the control, following enzymatic hydrolysis with or without ultrafiltration. In particular, the content of Gly, Pro, and Glu was much higher in CH-Alcalase < 3 kDa (Gly 218 mg/g, Pro 152 mg/g, and Glu 120 mg/g) than CH-Alcalase (Gly 149 mg/g, Pro 95 mg/g, and Glu 78 mg/g). An increase in the content of these amino acids is strongly related to enhanced antioxidant capabilities [16,20]. Gly and Pro contain hydrophobic amino acid groups, and Glu contains negatively charged amino acid groups. These amino acids have been reported to enhance antioxidant activity because of their increased solubilities in lipids or via free radical reactions [1,24].

### 2.3. Antioxidant and Anti-Aging Activities of Collagen Hydrolysates

Antioxidant activities of the CHs were measured using 2,2′-azino-bis-(3-ethylbenzothiazoline-6-sulfonic acid) (ABTS) radical-scavenging activity assay and reducing power assay. The ABTS radical-scavenging activity of the control (collagen suspension), CH-Alcalase, and CH-Alcalase < 3 kDa at different concentrations is shown (Figure 5A). ABTS radical scavenging activity of peptides assay is important to exclusively measure the ability of an antioxidant peptide to induce a hydrogen atom transfer [25]. ABTS radical-scavenging effects of all treatments increased in a concentration-dependent manner (*p* < 0.05). CH-Alcalase and CH-Alcalase < 3 kDa showed much higher ABTS radical-scavenging activity values, with 41.4%–88.2% compared to the control (<8.5%). Both CH-Alcalase and CH-Alcalase < 3 kDa had high ABTS radical-scavenging abilities of ~60% when concentrations were greater than 2.5 mg/mL. CH-Alcalase < 3 kDa showed a significantly higher ABTS radical-scavenging activity value than CH-Alcalase. In general, the antioxidant activity of peptides may be influenced by their amino acid sequences, the amount of free amino acids present, and the degree of hydrolysis and the molecular weight of peptides [26,27]. In particular, low molecular weight hydrolysates possessed stronger antioxidant properties compared to high molecular weight hydrolysates [2,26].

The reducing powers of the control, CH-Alcalase, and CH-Alcalase < 3 kDa are shown (Figure 5B). The reducing power of peptides may also serve as a significant indicator of its antioxidant potential [28,29]. The reducing power of CHs ranged from 0.074 to 0.424 in a dose-dependent manner (*p* < 0.05). The control group had the lowest reducing power, which did not change significantly with increasing concentrations of the control. In contrast to ABTS radical-scavenging activity, reducing power of CH-Alcalase was higher than that of CH-Alcalase < 3 kDa. Similar observations suggested that crude collagen hydrolysate may be more effective in reducing power than ultrafiltrated collagen peptide [30]. 

Inhibition of tyrosinase, collagenase, and elastase activity was used to verify their in vitro anti-aging effects [20]. The results of the inhibition of tyrosinase, collagenase, and elastase activity of the control, CH-Alcalase, and CH-Alcalase < 3 kDa are summarized (Table 3). Tyrosinase, collagenase, and elastase inhibitors have been used as important ingredients of cosmetics for skin whitening, anti-aging and anti-wrinkling, respectively. Collagenase and elastase, especially, are known to be major enzymes responsible for dehydration and wrinkle formation on the skin surface [31]. The results indicated that the tyrosinase inhibition effect of vitamin C (95.50%, 1 mg/mL, positive control) was higher than that of the other treatments (Table 4). Tyrosinase inhibition effects of control, CH-Alcalase, and CH-Alcalase < 3 kDa groups were 28.21%, 15.44%, and 30.20%, respectively. Thus, the CHs did not show a better skin whitening effect compared to the control. Collagenase inhibition by the control, CH-Alcalase, and CH-Alcalase < 3 kDa groups were 6.45%, 54.37%, and 61.90%, respectively. In addition, CH-Alcalase and CH-Alcalase < 3 kDa inhibition of collagenase corresponded to that of vitamin C at 1 mg/mL. Thus, CHs obtained from this study may be effective collagenase inhibitors that may possibly play an important role in anti-aging activities. However, all collagen samples did not display elastase inhibition effects, which may result from a lack of skin elasticity. Overall, vitamin C (1 mg/mL) inhibited various activities of the enzymes in the following order: tyrosinase (95.50%) > collagenase (48.09%) > elastase (27.00%). CH-Alcalase (5 mg/mL) inhibited various activities of the enzymes in the following order: collagenase (54.37%) > tyrosinase (15.44%) > elastase (no activity). CH-Alcalase < 3 kDa (5 mg/mL) inhibited these activities in the following order: collagenase (61.90%) > tyrosinase (30.20%) > elastase (no activity). Similar trends reported in another study indicated that collagen hydrolysates showed a potential to act as anti-aging agents and collagenase inhibitors [20].

## 3. Materials and Methods 

### 3.1. Porcine Skin Pretreatment

Porcine skin was purchased from a local supplier (Seoul, Korea), and all visible fat and connective tissues of the porcine skin were removed using a razor blade. The porcine skin used in this study was obtained from one porcine in order to minimize biological variation. Trimmed porcine skin was washed in water at 90 °C for 1 min four times to remove fat and residual materials. The skin was then cut into 1 cm square sections and pulverized in distilled water for 3 min using a four-wing blade blender (CNHR-26, Bosch, Hong Kong, China). The pulverized porcine skin was homogenized at high-speed (25,000 rpm) for 5 min using an Ultra Turrax^®^ (T25, IKA Labotechnik, Staufen, Germany). Approximately 100 g of the porcine skin mixture (50% final solid contents) was vacuum-packaged and frozen at −20 °C and stored for use within 1 month. 

### 3.2. Commercial Proteases and Reagents

Alcalase, Flavorzyme, Neutrase, and Protamex were purchased from Novozymes (Bagsvaerd, Denmark). Bromeline and Papain were purchased from Daesong Sangsa (Seoul, Korea). All chemicals for antioxidant and anti-aging tests were purchased from the Sigma-Aldrich Chemical Company (St. Louis, MS, USA). All other reagents and solvents used in this study were of analytical grade.

### 3.3. Enzyme Hydrolysis

Enzymatic hydrolysis was developed for the respective food-grade commercial enzymes used (based on the manufacturer’s recommendations; Table 4). The prepared porcine skin mixture was diluted with distilled water to a final solid content of 5%. This concentration was selected to ensure flow behavior because of its low viscosity. The 5% porcine skin mixture was termed the collagen suspension (or control). The collagen suspension was hydrolyzed in reactors using six food-grade commercial enzymes at an enzyme:substrate ratio of 1:100. Sample aliquots (5 mL) were drawn at 1, 3, 6, 12, and 24 h of hydrolysis and immediately heated at 100 °C for 10 min to inactivate the enzyme, followed by cooling to 0 °C using ice water. During the time of sampling, pH was controlled using 1 M NaOH as appropriate. After cooling, samples were centrifuged at 4000× *g* for 15 min, and the supernatant (collagen hydrolysate; CH) was collected. CH was concentrated further by ultrafiltration, using an Amicon^®^Stirred Cells system (Catalog No. UFSC 20001, EMD Millipore Corporation, Burlington, MA, USA) with a 3 kDa molecular weight cut-off (MWCO) (Ultracel^®^Membrane, EMD Millipore Corporation, Burlington, MA, USA) at 60 psi nitrogen gas, at 20 °C. The prepared CH was freeze-dried and kept in air-tight containers at 20 °C until analysis. 

### 3.4. Determination of pH, Protein Recovery, Solubility Free Amino Group Content, and Production Yield 

The pH of the samples (control and CHs) was determined using a pH meter (Model S220, Mettler Toledo GmbH, Columbus, OH, USA). The protein recovery or solubility was determined by estimating protein content using bicinchoninic acid (BCA) protein assay, according to the manufacturer’s instructions (Sigma-Aldrich, St. Louis, MS, USA) with serum albumin as the standard. Free amino group content was determined via a 2,4,6-trinitrobenzene sulfonic acid (TNBSA) assay according to the manufacturer’s instructions (Thermo Fisher Scientific, Waltham, MA, USA) using L-leucine as the standard. Both wet and dry weights were measured to calculate the production yield.

### 3.5. Molecular Weight Distribution

#### 3.5.1. Sodium Dodecyl Sulfate-Polyacrylamide Gel Electrophoresis (SDS-PAGE)

The SDS-PAGE patterns of samples (control and CHs) was measured according to a previously reported method [10]. Samples were diluted with 8 M urea (final protein concentration, 4 mg/mL). Each sample was mixed with one part of KTG 020 sample buffer (10% of glycerol, 2% of SDS, 0.003% of bromophenol blue, 5% of β-mercaptoethanol, and 63 mM Tris-HCl, pH 6.8) from KOMA Biotech Inc., (Seoul, Korea), and boiled for 2 min. The sample mixture (20 μL) was loaded into EzWay^TM^ PAG 6% acrylamide gels (KOMA Biotech Inc., Seoul, Korea). Following electrophoresis, the gel was fixed, stained, and de-stained. The molecular weights were determined using wide-range molecular weight standards between 10 and 210 kDa.

#### 3.5.2. Gel Permeation Chromatography (GPC)

The molecular weight distribution of samples was determined according to a previously reported method [3]. Gel permeation chromatography (GPC) was performed using a YL 9100 high-performance liquid chromatography (HPLC) system (YL 9100, Younglin Instrument Co., Ltd, Gyeonggi-do, Korea) equipped with three Ultrahydrogel TM 120 columns (7.8 × 3000 mm) from Waters (Milford, MA, USA). The mobile phase was distilled/deionized water at a flow rate of 1.0 mL/min, and the molecular weight distributions of the collagen peptides were monitored using a YL 9100 refractive index detector at 40 °C. A molecular weight standards kit (106–67,500 Da, Polymer Standards Service, Mainz, Germany) served as the standard.

### 3.6. Amino Acid Composition

The amino acid composition of the samples was analyzed through derivatization with 9-fluorenylmethoxycarbonyl (FMOC)-chloride and o-phthaldialdehyde (OPA) on an Ultimate 3000 HPLC system (Dionex, Idstein, Germany) equipped with two detectors (a fluorescence detector and a UV detector) and a VDSpher 100 C18-E (4.6 mm × 150 mm, 3.5 μm particle size, VDS Optilab, Berlin, Germany). The injection volume was 1.0 μL, and the mobile phase was composed of two eluents: a 40 mM sodium phosphate dibasic (pH 7) and a 45% (*v*/*v*) acetonitrile/45% (*v*/*v*) methanol solution. By connecting a UV detector and fluorescence detector, ultraviolet rays were detected at 338 nm, OPA derivative was detected at 450 nm of an emission wavelength and 340 nm of an excitation wavelength, FMOC derivative was detected at 305 nm of an emission wavelength of and 266 nm of an excitation wavelength. An amino acid mix (1.0 nmol mL^−1^ for each amino acid) was used for calibration.
(1)$$\mathrm{Amino}\;\mathrm{acid}\;\mathrm{content}\;\mathrm{in}\;\mathrm{CH}\;\left(\%\right)=\frac{A_{\mathrm{Total}\;\mathrm{amino}\;\mathrm{acid}}-A_{\mathrm{Free}\;\mathrm{amino}\;\mathrm{acid}}}{A_{\mathrm{Total}\;\mathrm{amino}\;\mathrm{acid}}}\times 100$$
where A_total amino acid_ was the content after hydrolysis using 6 M HCl (at 130 °C for 24 h), and A_free amino acid_ was the content after solubilizing in distilled water.

### 3.7. Evaluation of Antioxidant Activity

#### 3.7.1. 2,2′-Azino-bis-(3-ethylbenzothiazoline-6-sulfonic acid) (ABTS) Radical-Scavenging Activity

The ABTS radical-scavenging activity of CH was measured according to a previously reported method [20]. The ABTS radical cation was generated by mixing ABTS stock solution (7.0 mM) with potassium persulfate (2.45 mM) and incubating the resultant mixture in the dark at room temperature overnight. The ABTS radical solution was diluted in 5.0 mM phosphate-buffered saline (pH 7.4) to an absorbance level of 0.70 ± 0.02 at 734 nm. One mL of the diluted ABTS radical solution was mixed with 1 mL of each sample. Ten minutes later, sample (A_sample_, with sample) and control (A_control_, without sample) absorbances were measured at 734 nm. The ABTS radical-scavenging activity (%) was calculated as:
(2)$$\mathrm{ABTS}\;\mathrm{scavenging}\;\left(\%\right)=\frac{A_{\mathrm{Control}}-A_{\mathrm{Sample}}}{A_{\mathrm{Control}}}\times 100$$

#### 3.7.2. Reducing Power

The reducing power of CH was measured according to a previously reported method [20]. One mL of each sample was mixed with 1 mL of 0.2 M phosphate buffer (pH 6.6) and 1 mL of 1% potassium ferricyanide. The mixture was incubated at 50 °C for 20 min, and then 1 mL of 10% trichloroacetic acid was added. An aliquot of 2 mL from this incubation mixture was mixed with 2 mL of distilled water and 0.4 mL of 0.1% ferric chloride. After 10 min, the absorbance of the resulting solution was measured at 700 nm on a spectrophotometer (OPTIZEN, Mecasys Co., Daejeon, Korea) Increased absorbance (at 700 nm) of the reaction mixture was considered to indicate increased reducing power.

### 3.8. Evaluation of the Anti-Aging Effect

#### 3.8.1. Inhibition of Tyrosinase Activity

Tyrosinase inhibition was determined by a previously described method [20]. A premixture solution containing 70 μL of 0.1 M phosphate buffer (pH 6.8), 30 μL of mushroom tyrosinase (167 U/mL; Sigma-Aldrich, USA), and 20 μL of the sample was incubated for 5 min at 30 °C. Approximately 100 μL of 3,4-dihydroxyphenyl-l-alanine (l-DOPA) was then added to initiate the enzymatic reaction. Absorbance at 492 nm was measured for 20 min to monitor l-DOPA formation. Ascorbic acid (1 mg/mL) served as a positive control, which was used for comparison. The inhibition ratio was calculated as follows:
(3)$$\mathrm{Tyrosinase}\;\mathrm{inhibiton}\;\left(\%\right)=\frac{\left[\left(A-B\right)-\left(C-D\right)\right]}{\left(A-B\right)}\times 100$$
where A was a mixture with tyrosinase without sample; B was a mixture without sample and tyrosinase; C was a mixture with sample and tyrosinase; and D was a mixture with sample but without tyrosinase.

#### 3.8.2. Inhibition of Collagenase Activity

Collagenase inhibition was determined by a previously described method [20]. Briefly, 50 mM tricine buffer (pH 7.5) containing 10 mM calcium chloride and 400 mM sodium chloride was prepared. Then, 50 mL of a 1.0 mM *N*-[3-(2-furyl) acryloyl]-Leu–Gly–Pro–Ala solution and 0.2 mg/mL collagenase (from *Clostridium histolyticum*, Type IA, 0.5–50 FALGPA U/mg solid; Sigma-Aldrich, USA) were added in the presence and absence of samples. The reaction was stopped by adding citric acid (6%). The reaction mixture was separated by adding ethyl acetate. The absorbance of the supernatant was measured at 345 nm. Ascorbic acid (1 mg/mL) served as the positive control and was used for comparison. The percentage of inhibition was calculated as:
(4)$$\mathrm{Collagenase}\;\mathrm{inhibiton}\;\left(\%\right)=\frac{\left[\left(A-B\right)-\left(C-D\right)\right]}{\left(A-B\right)}\times 100$$
where A was a mixture with collagenase without sample; B was a mixture without sample and collagenase; C was a mixture with sample and collagenase; and D was a mixture with sample but without collagenase.

#### 3.8.3. Inhibition of Elastase Activity

Elastase inhibition was determined by a previously described method [20]. *N*-succinyl–Ala–Ala–Ala–p-nitroanilide (Suc–Ala–Ala–Ala–pNA) served as the substrate, and the release of *p*-nitroaniline was monitored for 20 min at 25 °C. A portion (1 μg) of type IV porcine pancreatic elastase (PPE) was dissolved in 1 mL of 0.2 M Tris-HCl buffer (pH 8.0). The reaction mixture contained 0.2 M Tris-HCl buffer (pH 8.0), 1 ppm PPE, 0.8 mM Suc–Ala–Ala–Ala–pNA, the sample, and the aforementioned substrate. Absorbance at 214 nm was measured. Ascorbic acid (1 mg/mL) served as the positive control used for comparison. The inhibition ratio was calculated as follows:
(5)$$\mathrm{Elastase}\;\mathrm{inhibiton}\;\left(\%\right)=\frac{\left[\left(A-B\right)-\left(C-D\right)\right]}{\left(A-B\right)}\times 100$$
where A was a mixture with elastase and without sample; B was a mixture without sample and elastase; C was a mixture with sample and elastase; and D was a mixture with sample but without elastase.

### 3.9. Statistical Analysis 

Data are presented as the mean ± standard deviation (SD). The significance of differences between groups was assessed using multiple comparisons and analysis of variance (ANOVA), followed by the Tukey honest significant difference (HSD) test. Differences with *p* values of less than 0.05 were considered statistically significant. 

## 4. Conclusions

In this study, collagen hydrolysates, a functional food ingredient, were successfully produced through enzymatic hydrolysis, followed by ultrafiltration and purification. Various commercial proteases were tested for their potential usefulness in manufacturing proper collagen hydrolysates. CHs hydrolyzed by Alcalase were the most effectively hydrolyzed CHs, and ultrafiltration followed by purification was effective in generating active peptides with low molecular weights. Results showed that the CHs displayed excellent antioxidant and collagenase inhibition activities. Therefore, CHs obtained by this study may be used in food, cosmetics, or pharmaceutical industries as a natural additive, possessing anti-oxidative and anti-aging properties. Further studies involving an in vivo evaluation of the aging activities of active peptides in human skin may prove to be useful.

## Figures and Tables

**Figure 1 molecules-24-01104-f001:**
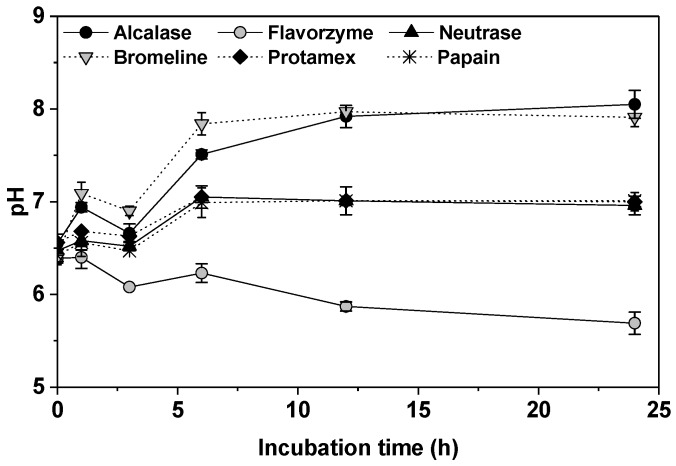
Change in pH of collagen hydrolysates depending on incubation time.

**Figure 2 molecules-24-01104-f002:**
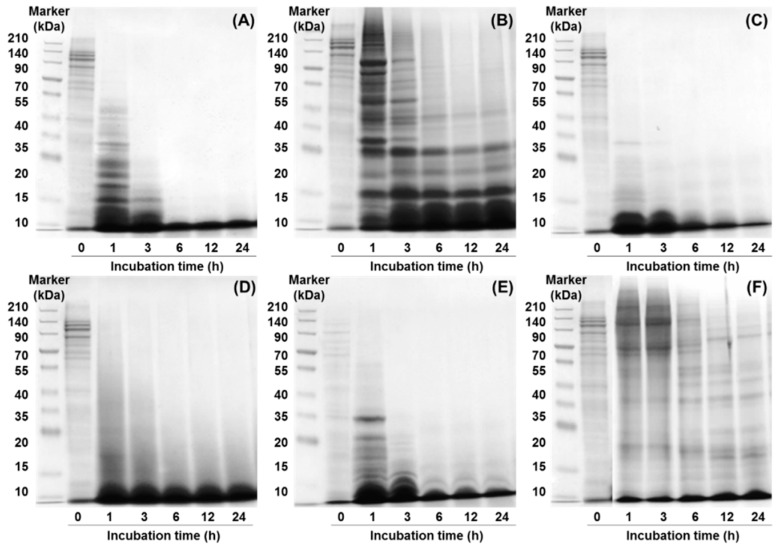
Change in the sodium dodecyl sulfate-polyacrylamide gel electrophoresis (SDS-PAGE) patterns of collagen hydrolysates hydrolyzed by different proteases: (**A**) Alcalase, (**B**) Flavorzyme, (**C**) Neutrase, (**D**) Bromeline, (**E**) Protamex, and (**F**) Papain.

**Figure 3 molecules-24-01104-f003:**
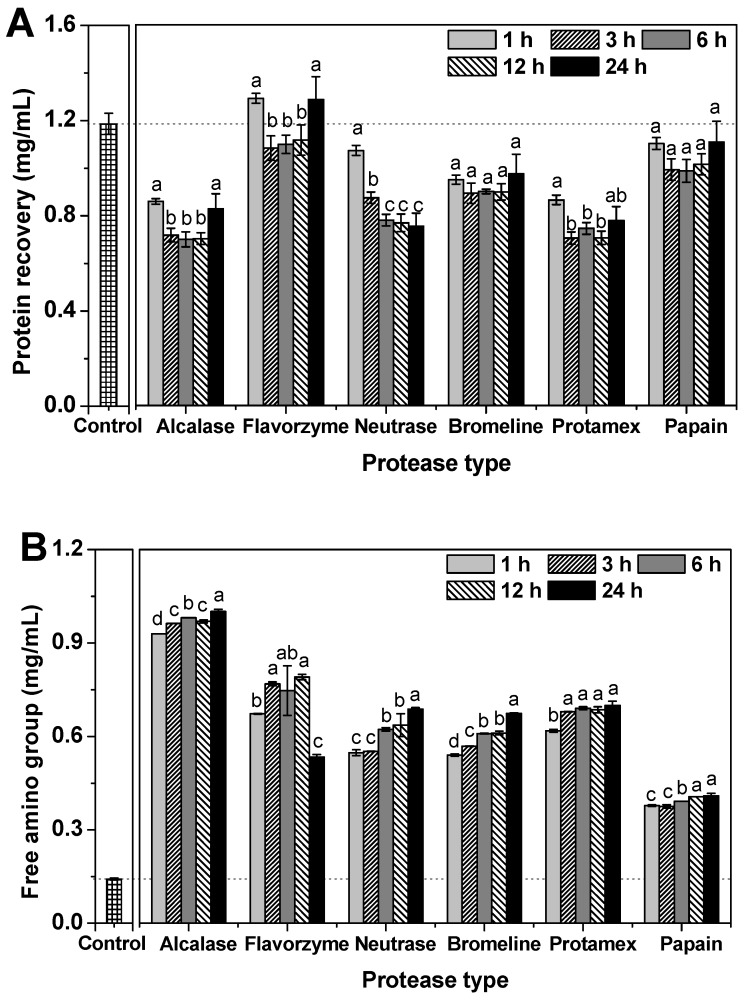
Change in protein recovery (**A**) and free amino groups (**B**) of collagen hydrolysates (CH) depending on incubation time. Data expressed as mean ± standard deviation (SD). Data denoted by different letters (a–d) show statistically significant differences depending on incubation time (*p* < 0.05).

**Figure 4 molecules-24-01104-f004:**
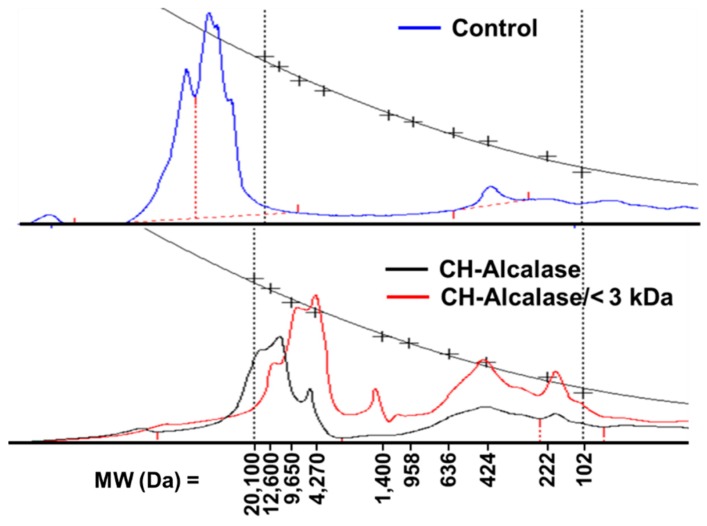
Change in molecular weight distribution of collagen hydrolysates (CH).

**Figure 5 molecules-24-01104-f005:**
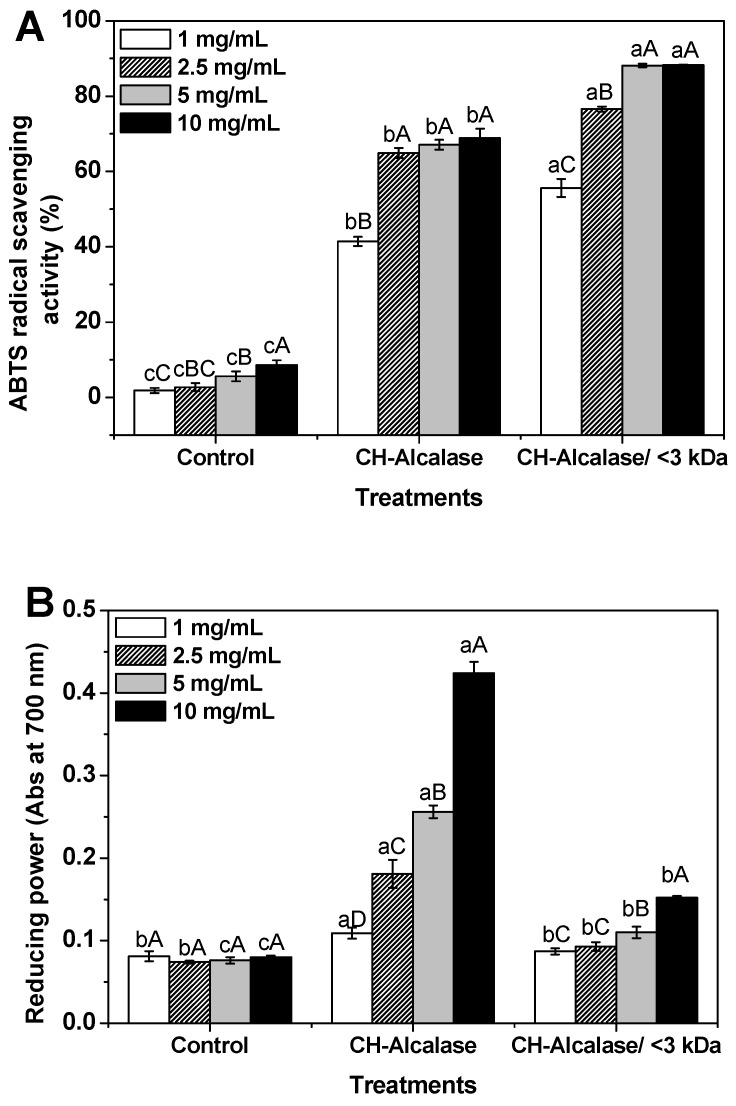
Antioxidant activities of collagen hydrolysates (CH) in 2,2′-azino-bis-(3-ethylbenzothiazoline-6-sulfonic acid) (ABTS) radical scavenging (**A**) and reducing power (**B**) assays. Data denoted by different letters (a–c) show statistically significant differences depending on different treatments (*p* < 0.05). Data denoted by different letters (A–D) show statistically significant differences depending on concentration (*p* < 0.05).

**Table 1 molecules-24-01104-t001:** Solubility, free amino acid group content, and yield of collagen hydrolysates (CH) after lyophilization.

Treatment	Solubility %	Free Amino Group %	Yield % ^1^
Control	11.95 ± 0.62 ^b,^*	0.79 ± 0.06 ^c^	27.27
CH-Alcalase	19.83 ± 0.04 ^a^	5.69 ± 0.02 ^b^	21.76
CH-Alcalase/<3 kDa	21.17 ± 0.15 ^a^	14.17 ± 0.25 ^a^	12.42

^1^ Powder production yield against raw porcine skin; * Values denoted by different letters (a–c) indicate statistically significant differences (*p* < 0.05).

**Table 2 molecules-24-01104-t002:** Change in the amino acid composition of collagen hydrolysates (CH).

Amino Acids	Control	CH-Alcalase	CH-Alcalase/<3 kDa
Aspartic acid	34.53	43.09	63.74
Glutamic acid	62.21	78.38	120.51
Asparagine	N.D. *	N.D.	N.D.
Serine	19.75	24.42	35.56
Glutamine	N.D.	N.D.	N.D.
Histidine	5.30	6.03	8.05
Glycine	116.63	148.61	218.5
Threonine	10.50	13.83	19.32
Arginine	44.05	52.01	73.05
Alanine	46.90	59.02	91.17
GABA	N.D.	N.D.	N.D.
Tyrosine	5.96	6.49	9.30
Valine	16.22	19.73	29.85
Methionine	4.30	4.39	9.29
Phenylalanine	13.42	16.25	22.51
Isoleucine	9.00	10.69	14.76
Leucine	19.86	22.74	33.24
Lysine	18.09	21.44	29.90
Proline	91.14	95.09	152.85

Unit: mg/g * N.D.: Not detected.

**Table 3 molecules-24-01104-t003:** Anti-aging effect of collagen hydrolysates (CH) in tyrosinase, collagenase, and elastase inhibition activity.

Treatments ^1^	Tyrosinase Inhibition Activity (%)	Collagenase Inhibition Activity (%)	Elastase Inhibition Activity (%)
Ascorbic acid ^2^	95.50 ± 0.02 ^a,^*	48.09 ± 0.01 ^b,c^	27.00 ± 0.05
Control	28.2 ± 2.8 ^b^	6.45 ± 3.3 ^d^	-
CH-Alcalase	15.44 ± 0.01 ^c^	54.4 ± 9.0 ^b^	-
CH-Alcalase/<3 kDa	30.20 ± 0.05 ^b^	61.9 ± 2.6 ^a^	-

^1^ All collagen hydrolysates were dissolved at 5 mg/mL concentration; ^2^ Ascorbic was dissolved at 1 mg/mL concentration; * Data denoted by different letters (a–c) show statistically significant differences (*p* < 0.05).

**Table 4 molecules-24-01104-t004:** Characteristics of commercial proteases.

Name	Type	Origin	Optimal Temperature (°C)	Optimal pH
Alcalase	endo (serine endoprotease)	*Bacillus licheniformis*	60	8
Flavorzyme	exo (aminopeptidase), endo mix	*Aspergillus oryzae*	60	6.5
Neutrase	endo (metallo-endoprotease)	*Bacillus amyloliquefaciens*	45	8
Bromeline	-	Pineapple	60	7
Protamex	endo (serine endoprotease)	*Bacillus licheniformis Bacillus amyloliquefaciens*	60	7
Papain	-	Carica papaya	60	7

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
