# Peer review of "Anti-Oxidative and Anti-Aging Activities of Porcine By-Product Collagen Hydrolysates Produced by Commercial Proteases: Effect of Hydrolysis and Ultrafiltration"

_molecules, 2019, doi:10.3390/molecules24061104_

Round 1

Reviewer 1 Report

The manuscript is dealing with very interesting subject. However, I would suggest the authors to emphasize in the conclusion that only one sample was tested and additional samples should be tested as well, both porcine and other sources. 

Furthermore, typing errors should be checked, for example the title: ant-oxidative instead of anti-oxidative.

tables are not numbered by the order - you have table 1 as the last one appearing in the text.

in Materials and methods, section 3.6. Amino acid composition, line 314 you state that HPLC has 2 detectors - were they both used for identification? If not, this is irrelevant, state only the one that was used.

in Conclusions, line 393 - the last sentence is not finished, the text is missing.

Author Response

Reviewer 1: Comments and Suggestions for Authors

The manuscript is dealing with very interesting subject. However, I would suggest the authors to emphasize in the conclusion that only one sample was tested, and additional samples should be tested as well, both porcine and other sources. 

à We appreciate your time and effort for constructive review. In our study, only porcine skin of main body was used for sample. And, we selected “porcine skin hydrolysates (collagen hydrolysates, CH)” hydrolyzed by Alcalase enzyme, and determined fraction condition of 3 kDa (CH-Alcalase/< 3 kDa). So, comparison of CH-Alcalase and CH-Alcalase/< 3 kDa is to investigate fractionation effect.

Furthermore, typing errors should be checked, for example the title: ant-oxidative instead of anti-oxidative.

à Thank you for checking error. We corrected it. (Page 1)

tables are not numbered by the order - you have table 1 as the last one appearing in the text.

à Thank you for your detailed checking. We all corrected them.

in Materials and methods, section 3.6. Amino acid composition, line 314 you state that HPLC has 2 detectors - were they both used for identification? If not, this is irrelevant, state only the one that was used.

à We re-wrote this section 3.6. Amino acid composition.

à Line 300-312: The amino acid composition of the samples was analyzed through derivatisation with flourenylmethoxycarbonyl (FMOC)chloride and ophthaldialdehyde (OPA) on an Ultimate 3000 HPLC system (Dionex, Idstein, Germany) equipped with two detectors (a fluorescence detector and a UV detector) and a VDSpher 100 C18-E (4.6mm x 150mm, 3.5um/VDS optilab, Germany). Injection volume was 1.0 μL and the mobile phase was composed of two eluents: a 40 mM Sodium phosphate dibasic (pH 7); and a 45% (v/v) acetonitrile/45% (v/v) methanol solution. By connecting UV detector and fluorescence detector, ultraviolet rays were detected at 338 nm; emission wavelength of the OPA derivative at 450 nm, excitation wavelength at 340 nm, emission wavelength of the FMOC derivative at 305 nm and excitation wavelength at 266 nm. An amino acid mix (1.0 nmol mL−1 for each amino acid) was used for calibration.

in Conclusions, line 393 - the last sentence is not finished, the text is missing.

à Thank you for your detailed checking. We completed this sentence.

à Line 380-381: Further studies involving an in vivo evaluation of the aging activities of active peptides in human skin may prove to be useful.

Reviewer 2 Report

COMMENTS FOR THE AUTHORS

Line 25. I think that it is of vital importance to include in the introduction and in the material and methods section the species which you have been working with. In the introduction it is also important to include some information regarding the industry from which the skin has been obtained or from which it could be obtained, the amount of skin by-products that could be generated in such industry and that are potentially valorized, in order to know if there is enough raw material to work with, if such waste represents an environmental problem, etc.   

Line 27: Please consider to change “presently” for other word in order to clarify the sentence.

Line 28: Please include an “a” between “is” and “high”, and remove the “a” just before “fibrous”.

Line 56: Please uniform the use of z or s in the words “hydrolyze” and “hydrolysate”.

Line 57: Please revise the grammar of the sentence especially regarding the use of “producing”.

Line 66: In general, the results section lacks of a comparative discussion regarding other results found using porcine collagen. Please revise and include some information accordingly.

Line 67: I would suggest modify the title of the heading to clarify the meaning.

Line 68: I would suggest to use subheadings to introduce each paragraph (change in pH;SDS-PAGE; protein recovery; free amino acid).

Line 72: What it is “the collagen suspension”? Please specify anywhere. It is not easy to find such information in the manuscript.

Lines 72-75: I do not understand why the pH increases from 6.39 or 6.55 to pH 8.0 in such a long time as 8 hours of incubation process for Alcalase hydrolysis or 12 hours hydrolysis in Flavourzyme hydrolysis? Why authors do not increases the pH of the medium up to the optimal pH of each enzyme just before starting the hydrolysis process? It is not specified in the material and methods section.

Line 81: As indicated before, I would suggest introducing a subheading section for SDS discussion.

Line 82: What is the “control”? If stated somewhere in the manuscript, I could not easily find the explanation of in which it is consisted of.

Line 83: from the figure 2 I would also understand a “rapid” hydrolysis in Bromeline hydrolysis, as there are not any bands after 1 hour of hydrolysis. Please clarify it.

Line 85: I understand that CH-Bromeline did not vary with incubation time because it has been completely hydrolysed after just 1 hour. Please clarify.

Lines 86-87: The bands also completely disappear after 6 hours in the Neutrase hydrolysate (as well as in the Alcalase hydrolysate).Please clarify. I would also mention that collagen is also a good substrate for some of the other enzymes tested. Please clarify or modify.

Lines 91-99Authors provide information of endo-exo activity for Alcalase and Flavorzyme, what about the other enzymes employed?

Line 101-102It should be included in the SDS-PAGE a molecular weight Collagen marker.

Line 103: As stated before, I would suggest include a subheading section for Protein recovery discussion.

Line 104: Again, what does “collagen suspension”consists of? It is not easy to find such information throughout the manuscript. Please revise and modify it throughout the manuscript. And please clarify throughout the manuscript if “collagen suspension”, “control” and “pretreatment sample” are the same. If so, please unify.

Line 112: Please revise the grammar of the sentence, especially regarding the use of “In estimating”.

Lines 119-121: I would suggest rewriting this sentence because the optimum hydrolysis time it depends on several factors such as the species selected, the raw material, the enzyme used, the raw material/enzyme rate, the concentration of the enzyme, the degree of hydrolysis pursued, etc. Therefore it should not be generalized that the time should be of 24 hours.

Line 123: Please, change “Figure B3” for “Figure 3B”.

Line 139: I would suggest modifying the heading because it is confuse. For example: “Ultrafiltration effect on collagen hydrolysate properties”.

Lines 145-147: Please revise the English grammar: “following enzymatic hydrolysis” and “after hydrolysis” is redundant.

Line 161: Only the relative molecular weight of CH-Alcalase is presented in figure 4, what about the other hydrolysates?

Line 175: Only the amino acid composition of CH-Alcalase is presented in table 3. What about the other hydrolysates?

Lines 177-178: Please could you explain the reason why the amount of amino acid is higher in hydrolysates (ultrafiltrated or not) than that of the control?

Line 187: Please remove the sentence “CH-Alcalase was hydrolyzed by Alcalase” because it is redundant and it should have been already explained in material and methods section.

Line 251: The same than above comment.

Line 252: Such explanation is again obvious and it has been already explained in M&M section.

Line 256: Please explain the reason why only “one porcine skin” has been employed for the experiments. Is it the whole complete skin of a unique individual? Or it is just a piece of it? In case it is only a piece of a unique individual skin it should be specified from which part of the animal it came from (ventral?dorsal?...). The reason why only one part of the skin has been selected for the study should also be included. The state of the raw material (skin) in the moment of which it has been purchased (frozen, fresh, etc.) In my opinion the sampling should include more than one skin from different individuals to avoid the bias of using only one skin.

Line 259: A collagen denaturation process could occur by heating it at 90°C?

Line 260; Instead “pulverized” should it be “homogenized”? Besides, why use water? Should it be better to remove the water before store the vacuum-packaged at -80°C.  A -20°C store temperature should not be enough?

Line 272: I suggest changing “optimized” by “developed”, because it has been not certainly an optimization study.

Lines 272-273: Please revise the grammar of the sentence, it is confusing.

Line 276: I suggest to modify “baths” for “reactors”.

Line 280: it is not clear the reason why HCl is employed. I understand that if the initial pH is about pH 6.5, the NaOH is necessary to increase the pH up to the optimal pH of the Alcalase for example, which is 8. But it is not clear the use of HCl. Please revise and modify the section accordingly.

Line 285: Is it necessary to get a temperature as low as -80°C? Please explain.

Line 287: Please clarify in which “samples” the pH was determined.

Line 296: Please explain the reason for using urea.

Line 306-307: What are the differences between the three columns used? Please specify.

Line 308: Distilled or deionized water used?

Line 393: The sentence “Further studies involving an in vivo” seems to be incomplete.

Line 401: The reference list includes more literature regarding marine or plant collagen characterization/ extraction than collagen from porcine or other mammalian sources. Is it because there are more published studies regarding marine collagen? I would suggest including some other studies of porcine collagen characterization in order to enhance the discussion section.

Author Response

Reviewer 2: Comments and Suggestions for Authors

Line 25. I think that it is of vital importance to include in the introduction and in the material and methods section the species which you have been working with. In the introduction it is also important to include some information regarding the industry from which the skin has been obtained or from which it could be obtained, the amount of skin by-products that could be generated in such industry and that are potentially valorized, in order to know if there is enough raw material to work with, if such waste represents an environmental problem, etc.   

Line 27: Please consider to change “presently” for other word in order to clarify the sentence.

à We omitted this word “Presently”.

Line 28: Please include an “a” between “is” and “high”, and remove the “a” just before “fibrous”.

à We modified this sentence: Collagen is a high molecular weight, fibrous protein of animal origin ~ (Line 27)

Line 56: Please uniform the use of z or s in the words “hydrolyze” and “hydrolysate”.

à We uniformly revised to “hydrolyse” and “hydrolysate”.

Line 57: Please revise the grammar of the sentence especially regarding the use of “producing”.

à We revised this sentence.

à Line 55-57: One study reported that collagen hydrolysates with antioxidant activity produced using enzymatical hydrolysis with hydrolyzed by pepsin and pancreatin from porcine skin gelatin [11].

Line 66: In general, the results section lacks of a comparative discussion regarding other results found using porcine collagen. Please revise and include some information accordingly.

à We additionally attached some references: 3, 10, 11, 23

Line 67: I would suggest modify the title of the heading to clarify the meaning.

à We all modified title of the heading and subheading in results and discussion section.

Line 68: I would suggest to use subheadings to introduce each paragraph (change in pH;SDS-PAGE; protein recovery; free amino acid).

à We all modified title of the heading and subheading in results and discussion section.

Line 72: What it is “the collagen suspension”? Please specify anywhere. It is not easy to find such information in the manuscript.

à The collagen suspension means 5% porcine skin mixture (Line 71): The section, 3.3. Enzyme hydrolysis is detailly presented: The prepared porcine skin mixture was diluted with distilled water to a final concentration of 5% for enzymatic hydrolysis and was termed the collagen suspension (Line 261-263).

Lines 72-75: I do not understand why the pH increases from 6.39 or 6.55 to pH 8.0 in such a long time as 8 hours of incubation process for Alcalase hydrolysis or 12 hours hydrolysis in Flavourzyme hydrolysis? Why authors do not increases the pH of the medium up to the optimal pH of each enzyme just before starting the hydrolysis process? It is not specified in the material and methods section.

à The pH of at 0 h is the pH of collagen suspension. We adjusted pH value before starting the hydrolysis process. But, during incubation periods, the pH of samples changed (decreased), continuously. So, whenever samples measured the pH at the time of sampling, and it was controlled using 1 M NaOH as appropriate. We described in M.M section. (Line 267)

Line 81: As indicated before, I would suggest introducing a subheading section for SDS discussion.

We added subheading title in Line 80.

Line 82: What is the “control”? If stated somewhere in the manuscript, I could not easily find the explanation of in which it is consisted of.

Control means “collagen suspension”. So, we modified control to collagen suspension.

Line 83: from the figure 2 I would also understand a “rapid” hydrolysis in Bromeline hydrolysis, as there are not any bands after 1 hour of hydrolysis. Please clarify it.

à Thank you for your comment. We revised this sentence

à Line 83-84: Results indicate rapid hydrolysis in CH- Alcalase, CH-Neutrase, CH-Bromeline, and CH- Protamex, as there are not any bands after 3 h incubation.

Line 85: I understand that CH-Bromeline did not vary with incubation time because it has been completely hydrolysed after just 1 hour. Please clarify.

à Thank you for your comment. We revised this sentence

à Line 86-87: However, CH-Bromeline did not vary with incubation time because it has been completely hydrolysed after just 1 h incubation..

Lines 86-87: The bands also completely disappear after 6 hours in the Neutrase hydrolysate (as well as in the Alcalase hydrolysate).Please clarify. I would also mention that collagen is also a good substrate for some of the other enzymes tested. Please clarify or modify.

à Thank you for your comment. We revised this sentence

à Line 87-89: Interestingly, the CHs completely disappeared when the sample reacted with Alcalase or Neutrase for 6 h, indicating that collagen acted as a suitable substrate for some of the other enzymes tested.

Lines 91-99.  Authors provide information of endo-exo activity for Alcalase and Flavorzyme, what about the other enzymes employed?

à We modified the sentence to “Since Alcalase, Neutrase and Protamex are an endopeptidase~~” (Line 94) and attached some references; 15,16.

Line 101-102.  It should be included in the SDS-PAGE a molecular weight Collagen marker.

à All samples at 0 h are collagen suspension before protein hydrolysis. We used collagen suspension for collagen marker.

Line 103: As stated before, I would suggest include a subheading section for Protein recovery discussion.

à We added subheading title in Line 104: 2.1.3. Protein recovery and free amino group content

Line 104: Again, what does “collagen suspension” consists of? It is not easy to find such information throughout the manuscript. Please revise and modify it throughout the manuscript. And please clarify throughout the manuscript if “collagen suspension”, “control” and “pretreatment sample” are the same. If so, please unify.

àThe pretreatment sample (section. 3.1) contains 50% final solid contents in porcine skin (50% porcine skin mixture. (Line 251)

àControl means 5% porcine skin mixture and terms of “collagen suspension (control)”. (Line 262-264)

Line 112: Please revise the grammar of the sentence, especially regarding the use of “In estimating”.

We modified this word in Line 115: For hydrolysis efficiency related to incubation time~~

à We modified the word as your comment (Line 114).

Lines 119-121: I would suggest rewriting this sentence because the optimum hydrolysis time it depends on several factors such as the species selected, the raw material, the enzyme used, the raw material/enzyme rate, the concentration of the enzyme, the degree of hydrolysis pursued, etc. Therefore it should not be generalized that the time should be of 24 hours.

à We revised the sentences and references your comments.

à Line 121-123: Some reports indicate that the optimum hydrolysis time depends on several factor such as the species selected, the raw material, the enzyme used, the raw material/enzyme rate, the concentration of the enzyme, the degree of hydrolysis pursued, etc. [16].

Line 123: Please, change “Figure B3” for “Figure 3B”.

à We changed to “Figure B3”. (Line 126)

Line 139: I would suggest modifying the heading because it is confuse. For example: “2.2. Ultrafiltration effect on collagen hydrolysate properties”.

à We modified the heading title as your comments (Line 142)

Lines 145-147: Please revise the English grammar: “following enzymatic hydrolysis” and “after hydrolysis” is redundant.

à We modified this sentence: Line 147-148

à After enzymatic hydrolysis and ultrafiltration with 3 kDa molecular weight cut-off,~~

Line 161: Only the relative molecular weight of CH-Alcalase is presented in figure 4, what about the other hydrolysates?

à In this study, we selected “CH” hydrolyzed by Alcalase enzyme (CH-Alcalase) which is one of the best protein hydrolysis effect, and determined fraction condition of 3 kDa (CH-Alcalase/< 3 kDa). So, comparison of CH-Alcalase and CH-Alcalase/< 3 kDa is to investigate fractionation effect.

Line 175: Only the amino acid composition of CH-Alcalase is presented in table 3. What about the other hydrolysates?

à It is also same reason as previous question on Line 161.

Lines 177-178: Please could you explain the reason why the amount of amino acid is higher in hydrolysates (ultrafiltrated or not) than that of the control?

à The amount of amino acid was calculated the equation (1).

à The amount of total amino acid was similar with all three samples. But the amount free amino acid increased with processes of protein hydrolysis and ultrafiltration. Therefore, the amount of amino acid in ultrafiltrated or not ultrafiltrated CH samples relatively higher than control. You can also confirm in Table 1.

Line 187: Please remove the sentence “CH-Alcalase was hydrolyzed by Alcalase” because it is redundant and it should have been already explained in material and methods section.

à We removed the sentence “CH-Alcalase was hydrolyzed by Alcalase” on Figure 4.

Line 251: The same than above comment.

à We removed the sentence.

Line 252: Such explanation is again obvious and it has been already explained in M&M section.

à Yes, we removed the repeated contents as your comment.

Line 256: Please explain the reason why only “one porcine skin” has been employed for the experiments. Is it the whole complete skin of a unique individual? Or it is just a piece of it? In case it is only a piece of a unique individual skin it should be specified from which part of the animal it came from (ventral?dorsal?..). The reason why only one part of the skin has been selected for the study should also be included. The state of the raw material (skin) in the moment of which it has been purchased (frozen, fresh, etc.) In my opinion the sampling should include more than one skin from different individuals to avoid the bias of using only one skin.

à The porcine skin used in this study was obtained from one fresh porcine and it randomly sampled from various body part to minimize biological variation. We modified this sentence to avoid confusion.

Line 244-246: Porcine skin was purchased from a local supplier (Seoul, Korea), and all visible fat and connective tissues of the porcine skin were removed using a razor blade. The porcine skin used in this study was obtained from one porcine in order to minimize biological variation.

Line 259: A collagen denaturation process could occur by heating it at 90°C?

à The process is to remove fat and residual materials. We have just shorter time (1 min) and several times.

Line 260; Instead “pulverized” should it be “homogenized”? Besides, why use water? Should it be better to remove the water before store the vacuum-packaged at -80°C.  A -20°C store temperature should not be enough?

à Yes, it is enough. We modified it as your comments. (Line 251)

Line 272: I suggest changing “optimized” by “developed”, because it has been not certainly an optimization study.

à Thank you. We modified it as your comments

Lines 272-273: Please revise the grammar of the sentence, it is confusing.

à We revised this sentence.

à Line 261-264: The prepared porcine skin mixture was diluted with distilled water to a final concentration of 5%. This concentration was selected to ensure flow behavior due to its low viscosity. The 5% porcine skin mixture was termed the collagen suspension (or control).

Line 276: I suggest to modify “baths” for “reactors”.

We changed to “reactors” (Line 264)

Line 280: it is not clear the reason why HCl is employed. I understand that if the initial pH is about pH 6.5, the NaOH is necessary to increase the pH up to the optimal pH of the Alcalase for example, which is 8. But it is not clear the use of HCl. Please revise and modify the section accordingly.

à Thank you for your detailed comment. We omitted the word,”HCl”.

Line 285: Is it necessary to get a temperature as low as -80°C? Please explain.

à Thank you for checking error. We revised to “air-tight container at 20oC” (Line 273).

Line 287: Please clarify in which “samples” the pH was determined.

à We revised this sentence: The pH of the samples (control and CHs) was determined using a pH meter. (Line 275)

Line 296: Please explain the reason for using urea.

à It is general method. We added the reference: The SDS-PAGE patterns of samples (control and CHs) was measured according to a previously reported method [10]. (Line 284-285)

Line 306-307: What are the differences between the three columns used? Please specify.

à We revised this section, 3.6. Amino acid composition, in Materials and Mathods.

à Line 300-312: The amino acid composition of the samples was analyzed through derivatisation with flourenylmethoxycarbonyl (FMOC)chloride and ophthaldialdehyde (OPA) on an Ultimate 3000 HPLC system (Dionex, Idstein, Germany) equipped with two detectors (a fluorescence detector and a UV detector) and a VDSpher 100 C18-E (4.6mm x 150mm, 3.5um/VDS optilab, Germany). Injection volume was 1.0 μL and the mobile phase was composed of two eluents: a 40 mM Sodium phosphate dibasic (pH 7); and a 45% (v/v) acetonitrile/45% (v/v) methanol solution. By connecting UV detector and fluorescence detector, ultraviolet rays were detected at 338 nm; emission wavelength of the OPA derivative at 450 nm, excitation wavelength at 340 nm, emission wavelength of the FMOC derivative at 305 nm and excitation wavelength at 266 nm. An amino acid mix (1.0 nmol mL−1 for each amino acid) was used for calibration.

Line 308: Distilled or deionized water used?

à We revised this section, 3.6. Amino acid composition, in Materials and Mathods.

Line 393: The sentence “Further studies involving an in vivo” seems to be incomplete.

à Thank you for your detailed checking. We completed this sentence.

à Line 380-381: Further studies involving an in vivo evaluation of the aging activities of active peptides in human skin may prove to be useful.

Line 401: The reference list includes more literature regarding marine or plant collagen characterization/ extraction than collagen from porcine or other mammalian sources. Is it because there are more published studies regarding marine collagen? I would suggest including some other studies of porcine collagen characterization in order to enhance the discussion section.

à Yes. Almost published studies are regard about marine collagen. But we attached more references; 3, 10, 11, 23.
